# Combining information from parental and personal experiences: Simple processes generate diverse outcomes

**Judy A. Stamps** [ID] [1]*, **Alison M. Bell** [2]

1 Department of Evolution and Ecology, University of California, Davis, Davis, California, United States of America, 2 Department of Evolution, Ecology, and Behavior, University of Illinois, Urbana-Champaign, Urbana, Illinois, United States of America

⊕ These authors contributed equally to this work.
* jastamps@ucdavis.edu

**Data Availability Statement:** The computer code used for the models is provided in the Supplemental Information (S4 and S5 Appendices).

## Abstract

Experiences of parents and/or offspring are often assumed to affect the development of trait values in offspring because they provide information about the external environment. However, it is currently unclear how information from parental and offspring experiences might jointly affect the information-states that provide the foundation for the offspring phenotypes observed in empirical studies of developmental plasticity in response to environmental cues. We analyze Bayesian models designed to mimic fully-factorial experimental studies of trans and within- generational plasticity (TWP), in which parents, offspring, both or neither are exposed to cues from predators, to determine how different durations of cue exposure for parents and offspring, the devaluation of information from parents or the degradation of information from parents would affect offspring estimates of environmental states related to risk of predation at the end of such experiments. We show that the effects of different cue durations, the devaluation of information from parents, and the degradation of information from parents on offspring estimates are all expected to vary as a function of interactions with two other key components of information-based models of TWP: parental priors and the relative cue reliability in the different treatments. Our results suggest empiricists should expect to observe considerable variation in the patterns observed in experimental studies of TWP based on simple principles of information-updating, without needing to invoke additional assumptions about costs, tradeoffs, development constraints, the fitness consequences of different trait values, or other factors.

## Introduction

The trait values which individuals develop over the course of ontogeny can be affected by their own experiences earlier in life (within-generational plasticity, WGP), by the experiences of their parents (transgenerational plasticity, TGP), or by the combined effects of both (trans and within- generational plasticity, TWP) [1–5]. It is often assumed that one reason why the

**Funding:** This material is partially based upon work supported by the National Science Foundation under Grant No. IOS 1121980 and the National Institutes of Health under award number 2R01GM082937-06A1.

**Competing interests:** The authors have declared that no competing interests exist.

experiences of parents and offspring might have adaptive effects on trait development is that those experiences provide information about conditions in the external environment that the offspring are likely to experience later in life [6–12]. In addition, information about conditions in the external environment can also be provided by genes, inherited epigenic factors and parental phenotypes [7, 13, 14]. Hence, in order to appreciate how information from an individual's distant and immediate ancestors and its own personal experiences might affect the development of traits that are the focus of studies of WGP, TGP and TWP, we must consider how information from a variety of different sources combines within and across generations to affect the information-state of that individual.

Bayesian updating provides a consistent and natural way to combine information from different sources and different times to estimate the value of variables in the external environment [15, 16]. As a result, in recent years researchers have begun to use Bayesian approaches to study how information from ancestors and personal experiences might combine over the course of ontogeny [17–23], review in [24]. These models assume that an individual's estimate of conditions in the external environment can change over the course of ontogeny, as its initial estimate when it is naïve, based on information provided by its ancestors (modelled by its naïve prior distribution, see List of Terms, S1 Appendix) is updated on the basis of information provided by its own experiences. Changes in an individual's estimate over time are assumed to drive developmental plasticity within generations (i.e., WGP), because an individual's estimate of conditions in the external environment (e.g. its estimate of predator density at that locality) is assumed to affect the trait values it develops in response to that estimate (e.g. its level of anti-predator behavior). By extension, Bayesian models of development predict that if different individuals with the same naïve prior distribution were exposed to different informative experiences over the course of ontogeny, they would develop different trait values in response to those experiences [17, 18, 21].

By definition, empirical studies of WGP, TGP and TWP focus on the development of phenotypic traits, not information-states. However, Bayesian models that are designed to predict variation in phenotypic traits necessarily include assumptions about many variables besides those which are required to predict changes in an organism's information-state in response to cues from the external environment. For instance, in order to predict the phenotypes that we would observe if organisms evolved the developmental programs favored by natural selection, Bayesian models of development have included assumptions about costs of sampling, developmental constraints, ecological factors (e.g. correlations across environments between food availability and risk of predation), the fitness consequences of expressing different trait values at different ages, and other factors unrelated to information-updating [17, 18, 21, 25]. As a result, in such models it is difficult to determine how information-updating per se contributes to their results. In addition, because of their complexity, these models typically rely on simple assumptions about information-updating, e.g., that there are only two discrete types of environment, that all cues are equally reliable, or that every individual is exposed to the cues for the same period of time. As a result, although such models demonstrate that information-updating can contribute to developmental plasticity, they are less suitable for investigating how different components of information-updating (e.g. different naïve priors, different cue reliabilities, or different durations of exposure to the same cues) interact to affect the information-states of individuals.

A different way to study how information-updating might affect developmental plasticity is to focus on information-states, instead of phenotypes [20, 22, 23]. Models which consider how information-states change within individuals and vary across individuals as a function of variation in prior distributions, cue reliability and other key components of information-based models have already provided insights into a number of questions about developmental

biology, including the ways that information-updating could contribute to variation across individuals or genotypes in their developmental trajectories [20, 22, 26], or variation among individuals or taxa in age-dependent plasticity and sensitive periods [23]. These analyzes indicate that empiricists might observe a variety of different patterns in empirical studies of these topics, simply based on the ways that we would expect information from different sources to combine over ontogeny within individuals. By extension, these results suggest that it might not be necessary to invoke assumptions about developmental or physiological constraints, tradeoffs, costs of sampling, the fitness consequences of trait values, or other factors to account for at least some of variation empiricists have observed among species, populations, or individuals in their developmental responses to the same experiences (e.g. [27–29].)

Recently Stamps and Bell [30] extended this approach by using simple Bayesian approaches to model experimental studies of developmental plasticity across and within generations (i.e., TWP) in response to informative cues. They found that interactions between two basic components of such models (the parent's prior distribution at the beginning of the study, and the relative reliability of the cues in the different treatment groups) had profound effects on the patterns of offspring information-states expected at the end of those experiments.

In the current study, we expand these analyses to consider three other variables that are likely to vary among experimental studies of TWP: 1) differences in the duration of exposure to the same cues for parents and offspring, 2) the extent to which information from parents is devalued, as compared to the information based on offspring experience, and 3) the extent to which information from parents is degraded as it is passed from parents to their offspring. We were particularly interested in whether and how variation in these three factors might interact with variation in the parental prior distributions and the relative reliability of the cues to affect the offspring information-states expected at the end of empirical studies of TWP.

As was the case in [30], we illustrate this approach by modelling fully factorial experiments of TWP in response to cues from predators, in which parents, offspring, both or neither are exposed to cues from predators, and then the trait values of the offspring are measured at the end of the experiment. Where P = exposed to cues from predators and N = not exposed to cues from predators, and where the first letter indicates the parental treatment and the second letter indicates the offspring treatment, the four treatment groups in this type of study consist of NN (neither parents and offspring exposed to cues), PN (parent exposed, offspring not exposed), NP (parent not exposed, offspring exposed) and PP (both parents and offspring exposed).

In our previous analyses, we assumed that parents and offspring were exposed to the same cues for the same period of time [30]. However, this assumption is often violated in experimental studies of TWP. For instance, of the 13 experimental studies of TWP in response to cues from predators listed in a recent review [31], the duration of exposure to the cues for parents and for their offspring differed in 11 of them. Hence, one important question is whether and how different durations of exposure to the same cues in parents and offspring might contribute to differences among the offspring in their information-states at the end of experimental studies of TWP.

In addition, in [30] we assumed that exposure to the same cues for the same period of time in parents and offspring would provide the same information to offspring. There are at least two reasons why this assumption need not be valid, and in particular, why the information based on a parent's experience might be less reliable than the information based on the same experience for their offspring. First, theory indicates that information based on parental experiences should be devalued, relative to information based on offspring experiences, if environmental conditions might change across or within generations [7, 32]. In nature, changes between generations in the value of a state of the environment can occur if offspring develop

in a different environment than their parents, e.g., as a result of natal dispersal in spatially heterogeneous environments [11, 14] or as a result of temporal shifts in environmental conditions that occur between generations [7, 11].

Along the same lines, if environmental conditions can change within generations, a parent's estimates of those conditions based on their experiences earlier in life might no longer be accurate by the time that they transmit information about these conditions to their offspring [7, 33]. This possibility is implicitly acknowledged by empiricists who study TWP when they use experimental protocols that minimize the time that elapses between parental exposure to the cues and the time when parents transfer information to their offspring (e.g. when parents are exposed to cues from predators as adults, just before offspring production) (e.g [34–36]).

Another reason why the information based on a parent's experience might be less reliable than the information based on an offspring's experience is because the information based on the parent's experience is degraded as a result of 'noise' in the extended information pathway by which parents to pass information to their offspring. The degradation of information from parents is likely because the transmission of information from parents to offspring requires additional proximate steps that are not required when offspring are directly exposed to the same cues [33]. For instance, parents that detect a cue from predators need to produce a signal (e.g., alter their parental behavior) based on their updated estimate of the chances that this predator lives nearby [37]. Then their offspring need to detect this signal and use it to update their own estimate of the probability that the predator is in residence. Since neither the parent's production of the signal nor their offspring's detection of that signal are likely to be error-free, the information based on the parent's experience could become less reliable by the time it reaches their offspring. In contrast to the devaluation of information provided by parents, which is assumed to be an adaptive response to spatial and/or temporal heterogeneity in environmental conditions, the degradation of the information provided by the parents is assumed to be an unavoidable consequence of the proximate mechanisms by which parents pass information based on their own experiences to their offspring.

## Material and methods

Details about the design and biological rationale for the models used as the basis for those in the current article have been provided in previous publications [20, 22, 23, 30]; see also the S1–S5 Appendices, and the S1 and S2 Figs). In brief, we assume that individuals are attempting to estimate the value of a state of the environment (e.g., predator density, or food abundance), and that this state can take on one of 100 possible exclusive values, which for convenience we allow to range from 0 to 1. We also assume that under natural conditions, the true value of the state is unlikely to change over a period equivalent to the duration of a TWP experiment (i.e., the period which begins when the parents are first exposed to the conditions in the P or the N treatments, and which ends when traits are measured in the offspring).

We assume that at the beginning of an experiment, parents begin with a prior distribution (Prior), which indicates their initial estimate of the probability, for each of the 100 possible values of the state, that each of those values is the true value of the state (see S1 and S2 Appendices). For instance, a parent might begin by assuming that the value of the state "predator density" is more likely to be low than it is to be intermediate or high. We assume that the parental Prior distribution is based on information from their ancestors (e.g., via genes, inherited epigenetic factors, grand-parental experiences), as well as any informative experiences the parents had before the onset of the experiment. We capitalize the parental Prior because it is the only prior that is specified by the user in the models; all subsequent priors (e.g., the offspring's prior when it is first exposed to the stimuli in the P or the N treatments) are generated

by Bayesian updating. The mean of the parental Prior indicates the parent's best point estimate of the value of the state before it is placed in the P or N treatments, and the variance of the parental Prior is inversely related to a parent's confidence in the accuracy of this initial estimate [20, 24]. We used the beta distribution to describe the shape of each parental Prior, because the beta distribution can describe distributions with many different shapes, and because the shape of any beta distribution can be easily specified using two parameters, $\alpha$ and $\beta$.

In preliminary analyses we analyzed parental Priors with a range of different means and variances, but here illustrate our major points using three parental Priors, with different means (0.1, 0.5 and 0.9), but the same variance (0.04). In other words, we assume that at the onset of the experiment, different parents might begin with different estimates of the value of the state (e.g., low (0.1), intermediate (0.5) or high (0.9)), but that all of them are equally confident that their initial estimate is correct. These three parental Priors are illustrated in the S1 Fig.

We assume that cues from the predator in the P treatment were always informative. That is, we assume that if an individual was exposed to a particular set of cues from a predator for a specific period in the P treatment, this experience would be more likely to occur for some values of the state than for others. For instance, if in the P treatment individuals were exposed to a high concentration of kairomones from the predator from birth to maturity, we assume that this experience would be most likely to occur if predator density was high, less likely to occur if predator density was intermediate, and unlikely to occur if predator density was low. Here, we use the term 'cumulative likelihood function' to describe, for all of the 100 possible values of the state, the conditional probability that a subject would have all of the experiences to which it was exposed in either the P or the N treatment, given each of the 100 possible values of the state. We used the beta distribution to describe the shapes of these cumulative likelihood functions, for the same reasons that we used it to describe different parental Priors (see above). The cumulative likelihood functions used for this study are illustrated in the S2 Fig.

In contrast to the situation for the conditions in the P treatment, which we assumed always provided information about the value of the state to the subjects, we analyzed two sets of models which differed with respect to their assumptions about the information provided by the N treatment. In the first set (N- models), the information provided by the N treatment was less reliable than the information provided by the P treatment. We analyzed this situation by assuming that the information provided by conditions in the P treatment was highly reliable, using a cumulative likelihood function with a shape modelled by $\alpha = 8$, $\beta = 1$. This likelihood function indicates that the conditions to which subjects were exposed in the P treatment would be much more likely to occur if the value of the state was high than if it was intermediate or low. In contrast, we assumed that the conditions in the N treatment provided no information about the state; in this case, the cumulative likelihood function for the N treatment had a uniform distribution ($\alpha = 1$, $\beta = 1$), indicating that the conditions in N would be equally likely to occur for any of the 100 possible values of the state.

In the second set of models (N* models), we assumed that the information provided by the P treatment and the information provided by the N treatment were equally reliable. In this case, the cumulative likelihood function for the N treatment was the mirror-image of the cumulative likelihood function for the P treatment. For instance, if the experiences in the P treatment resulted in a cumulative likelihood function that indicated with a high level of reliability that the value of the state of the environment was most likely to be high (e.g. likelihood modelled by a beta distribution with a shape indicated by $\alpha = 8$, $\beta = 1$), the conditions in the N treatment had a cumulative likelihood function that indicated with the same level of reliability that the value of the state was most likely to be low (e.g. likelihood with a shape modelled by $\alpha$

= 1, β = 8). In the latter case, the conditions to which the subjects were exposed in the N treatment would be much more likely to occur if the value of the state was low than if it was intermediate or high.

For each of the models, we specified the mean and variance of the parent's Prior distribution, then computed the mean and variance of the parent's posterior distribution (at the end of the parent's experiences in the P or the N treatment), and the mean and the variance of the offspring's posterior distribution (at the end of the offspring's experiences in the P or the N treatment) (see S2–S5 Appendices). In presenting the results, we focused on the mean of the offspring posterior distributions at the end of their respective treatments. The mean of an offspring's posterior distributions provides a simple way to describe the offspring's best point estimate of the value of the state at the end of the experiment. We focused on the means rather than the variances of the offspring posterior distributions because preliminary analyses indicated that by the end of the experiment, the variances of the offspring posterior distributions were typically low in all four treatment groups. For convenience, in this article, we use 'offspring estimate' to refer to the mean of an offspring's posterior distribution at the end of the experiment.

## Duration of exposure to the same cue in parents and offspring

Preliminary analyses showed that if parents and offspring were exposed to the same cues for the same period of time, differences between the parental and the offspring generation in the age of onset of the exposure period for each generation had no effects on the results. These results occur because in Bayesian updating models which assume that the true state of the environment is unlikely to change over time, if different subjects with the same prior distribution are exposed to the same cues, the order in which they were exposed to those cues has no effect on their final posterior distributions. Because the order-indifference of Bayesian updating is particularly relevant to analyses of sensitive periods and age-dependent plasticity, we defer discussion of this point to a study of that topic (Stamps, in prep.)

In contrast, preliminary analyses suggested that different durations of exposure to the same cues in parents and offspring would affect offspring estimates at the end of the experiment. In order to investigate the effects of different durations of exposure to the conditions P or N for parents and offspring on the results, we divided the total treatment period, T, for each generation into four intervals of equal length. For each interval, we specified one likelihood function if the subjects were exposed to the cue during that interval, and a different likelihood function if the subjects were not exposed to the cue during that interval. For instance, to model a situation in which parents were exposed to the cues for a longer period than the offspring, we assumed that parents in the P treatment were exposed to cues from the predator for all four intervals, but offspring in the P treatment were exposed to the cues from the predator for either one, two or three intervals, and were exposed to no cues from predators for the remaining interval(s). Similarly, to model a situation in which offspring were exposed to the cues for a longer period than the parents, we assumed that offspring in the P treatment were exposed to the cues for all four intervals, but parents in the P treatment were exposed to the cues for one, two or three intervals, and were exposed to no cues for the remaining intervals. In all of these models, the N treatment group was maintained with no cues from predators for all four intervals.

For each model, we specified the likelihood functions appropriate for the presence of cues in one interval and for the absence of cues in one interval (see below). Then, we began with a parental Prior, sequentially exposed the parents to the likelihood functions appropriate for each of their four intervals, sequentially exposed the offspring to the likelihood functions

appropriate for each of their four intervals, and then computed each offspring's posterior distribution at the end of these eight intervals.

As in [30], we computed separate models for the N- situation (in which the conditions during one time interval in the P treatment provided much more reliable information than the conditions during one interval in the N treatment) and for the N* situation (conditions during one interval in the P treatment provided information as reliable as the conditions during one interval in the N treatment). For the N- models, we assumed that no information was provided in an interval that lacked the cue. That is, the likelihood function for one interval spent in the absence of the cue had a shape indicated by a uniform distribution ($\alpha = 1$, $\beta = 1$). In the N* models, we assumed that the information provided by one interval spent in the absence of the cue was as reliable as the information provided by one interval spent in the presence of that cue. For instance, if the cumulative likelihood function for one interval in the presence of cues from a predator had a shape which conformed to the beta distribution generated by $\alpha = 2.5$, $\beta = 1$, the likelihood function for one interval in the absence of those cues had a shape which conformed to the beta distribution generated by $\alpha = 1$, $\beta = 2.5$ (see S2 Fig for examples of these likelihoods).

## Cues based on parental experience are devalued or degraded

Although the devaluation and the degradation of information based on parental experiences are assumed to occur as a result of different processes (see Introduction and Discussion), in both situations it is assumed that the information provided by the signal that parents pass along to their offspring based on their exposure to a given cue is less reliable than the information provided by the cue to which the parents had been exposed. In Bayesian terms, this means that although we would expect the likelihood function for the signal provided to their offspring by parents in the P treatment to have the same mean as the likelihood function for the cue in the P treatment for the offspring, we would expect the likelihood function for the parental signal to have a higher variance than the likelihood function for the cue for the offspring. Since we used beta distributions to describe the shape of the cumulative likelihood functions of both parents and offspring, this means that if we used a beta distribution to describe the likelihood function for the information provided to offspring by the conditions in the P treatment, we would use a different beta distribution with the same mean, but a higher variance, to describe the information provided by the parents to their offspring.

Here, we used the same procedure to model the devaluation of information from the parents and the degradation of information from the parents. That is, we assumed for both situations that the signal from the parent and the cue for the offspring had likelihood functions described by beta distributions with the same mean, but different variances. For instance, if the likelihood function for the P treatment for offspring had a shape modelled by a beta distribution with $\alpha = 8$, $\beta = 1$ (mean = 0.89, variance = 0.01), the likelihood function for the signal provided by parents to offspring by the parents in the P treatment might have a shape modelled by $\alpha = 3.5$, $\beta = 0.44$ (mean = 0.89, variance = 0.02). That is, we assumed that exposure to cues from predators in the P treatment yielded the same point estimate of the value of the state of the environment for parents and offspring (in this case, a relatively high value of 0.89, on a scale of 0 to 1), but that the reliability of the information provided by the parents to their offspring (indicated by the variance of the likelihood function) was lower than the reliability of the information provided by the same experience for the offspring (see S2 Fig for an illustration of this example).

In the N- models, the reliability of the information for the P treatments differed for parents and offspring, as was indicated above. However, given our assumption for these models that

the information provided by conditions in the N treatments was unreliable (see above), we assumed that the information provided by the N treatments was equally unreliable for both the parent and the offspring generation (modelled by $\alpha = 1$, $\beta = 1$).

In the N* models, in which conditions in the N treatment provided information as reliable as those in the P treatments, we devalued the information provided by parents in the N treatment groups to their offspring. For instance, if the likelihood function for the P treatment for the offspring had a shape modelled by $\alpha = 8$, $\beta = 1$, the likelihood function for conditions in the N treatment for offspring had a shape modelled by $\alpha = 1$, $\beta = 8$, but the likelihood function for the information provided by parents to their offspring had a shape modelled by $\alpha = 0.44$, $\beta = 3.5$.

All of the other assumptions for these models were the same as those for the baseline models described above.

## Results

In the 'baseline' models, in which parents and offspring were exposed to the same cues for the same period of time, and in which information from parents was neither devalued or degraded, the means of the offspring posterior distributions ("offspring estimates") varied as a function of interactions between the relative reliability of the cues in the different treatments and the parental Priors (Fig 1). In the N- models, in which the information provided by conditions in one treatment (here P, indicating the presence of cues from predators) was much more reliable than the information provided by the conditions in the other treatment (here, N, indicating the absence of cues from predators), the patterns observed varied as a function of the parental Priors. When the value of the state indicated by the parental Prior was very different from the value of the state indicated by the cues in the P treatment, we observed a 'jump-up' pattern, in which the value for the NN group was low, and the values for the other three groups (NP, PN, PP) were high and similar (but not identical) to each other. Since in this study we assumed that the cues in the P treatment indicated that the value of the state of the environment was high, we observed the jump-up pattern when the parental Prior strongly contradicted those cues (Prior mean = 0.1) (Fig 1A). In contrast, when the parental Prior and the cues in the P treatment both indicated similar values of the state (Prior mean = 0.9), the offspring estimates were similar for all four treatment groups. These results occurred because in these models, the conditions in the N treatment had no effect on an individual's information-state, so the patterns of offspring estimates were primarily affected by the 'discrepancy rule' of Bayesian updating (see [26] for more on this topic).

In the N* models, in which the information provided by conditions in the P treatment and the information provided by conditions in the N treatment were equally reliable, we observed a 'step-up' pattern, in which the offspring estimates for the NN treatment were low, those for the PP treatment were high, and those for the NP and PN treatments were intermediate. In this case the pattern did not vary as a function of the parental Prior: for the same set of parameter values, similar step-up patterns were observed for a range of parental Priors (Fig 1B). This result occurred because in the N* models, information about the value of the state of the environment was provided by the conditions in both treatment groups. After the offspring had been exposed to two doses of informative cues (based upon the parent's experience and their own personal experience), the initial estimate provided by the parental Prior no longer had much effect on their estimates of the value of the state of the environment. These results have been described and discussed in detail in [30], so we merely present examples of these patterns here for comparison with the results of the models in which we relaxed our assumptions about cue durations, devaluation, and degradation.

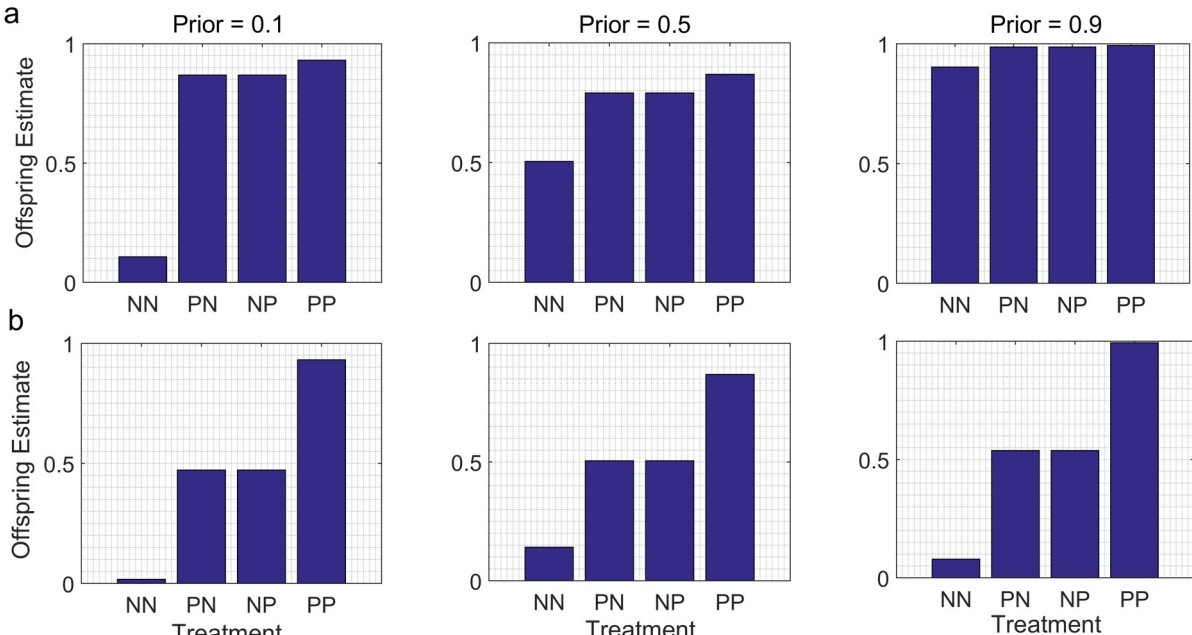

**Fig 1. How differences in the reliability of the cues in the different treatments affect offspring estimates of the value of conditions in the external environment.** P treatment = exposure to cues from a predator, N treatment = no cues from predator, first letter = parental treatment, second letter = offspring treatment. Predicted offspring estimates at the end of an experiment for each of four treatment groups (NN, PN, NP and PP) are indicated for parental Priors with three different means (0.1, 0.5, 0.9) and the same variance (0.04). a. N- models: conditions in the N treatment provide much less reliable information than conditions in the P treatment. In this example, information provided by conditions in the P treatment is modelled by a cumulative likelihood function with a shape indicated by α = 8, β = 1, indicating that the value of the state is likely to be high; information provided by conditions in the N treatment is modelled by a uniform distribution (α = 1, β = 1). b. N* models. Conditions in the N treatment provide information as reliable as conditions in the P treatment; the likelihood functions for the P and N treatments are mirror-images of one another. In this example, conditions in the P treatment indicate with a high level of reliability that the state of the environment is likely to be high (likelihood modelled by α = 8, β = 1); conditions in the N treatment indicates with equally high reliability that the state of the environment is likely to be low (likelihood modelled by α = 1, β = 8).

When parents and offspring were exposed to the same cues for different periods of time, the main effect was to generate differences between the offspring estimates for the PN and NP groups; these differences were not observed in the baseline models (compare Fig 1 with Figs 2 and 3). As intuition would suggest, the offspring estimates were higher for the treatment group which included the generation that had been exposed to the cues for a longer time. For instance, if the offspring in the P treatment group were exposed to the cues for a longer period than their parents, PN < NP (Fig 2A and 2B). Conversely, if the offspring in the P treatment group were exposed to the cues for a shorter time than their parents, then PN > NP (Fig 3A and 3B).

The overall patterns were otherwise similar to those found in the baseline models. That is, a jump-up pattern was detectable in the N- models when the parental Prior indicated a different value of the state than the cues in the P treatment, but not when the parental Prior indicated a value of the state similar to the value indicated by the cues in the P treatment (Figs 2A and 3A). In contrast, step-up patterns were detectable in the N* models for a range of parental Priors (Figs 2B and 3B). However, for comparable sets of parameter values (i.e., for the same parental Prior, and for the same likelihood function for the cues in the P treatment), the differences between the offspring estimates for the NP and PN groups were much less pronounced for the N- models than for the N* models (compare Fig 2A with Fig 2B, and Fig 3A with Fig 3B).

When information from the parents was devalued or degraded, the models predicted lower estimates of the value of the state for the PN group than for the NP group (Fig 4). The overall

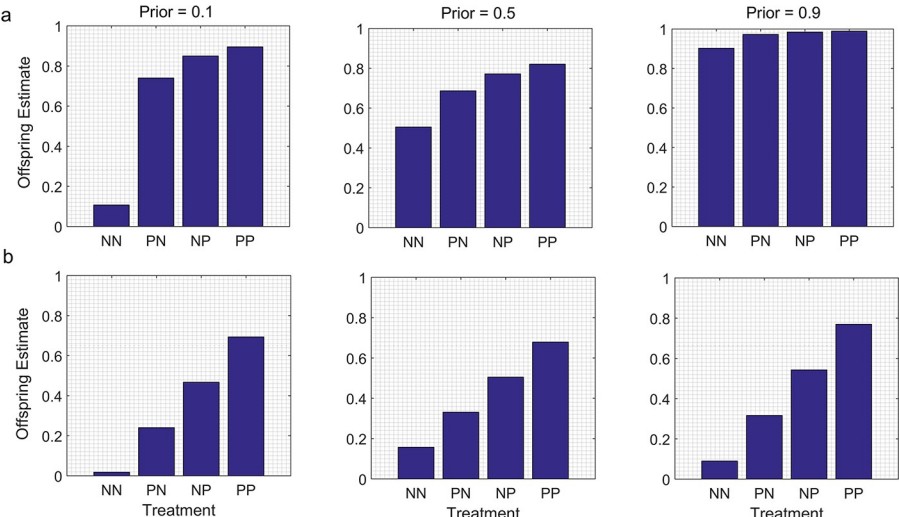

**Fig 2. Duration of exposure to the same cues is longer for offspring than for parents.** In the P treatment groups, parents are exposed to the presence of cues from predators for two of four time intervals, but offspring are exposed to cues from predators for all four time intervals (see text). a. N- models. In this example, the likelihood function for exposure to cues from a predator for one time interval has a shape indicated by α = 2.5, β = 1; the likelihood function for the absence of cues for one interval has a shape indicated by a uniform distribution (α = 1, β = 1). b. N* models. In this example, the likelihood function for exposure to cues from a predator for one interval has a shape indicated by α = 2.5, β = 1; the likelihood function for the absence of cues for one interval has a shape indicated by α = 1, β = 2.5.

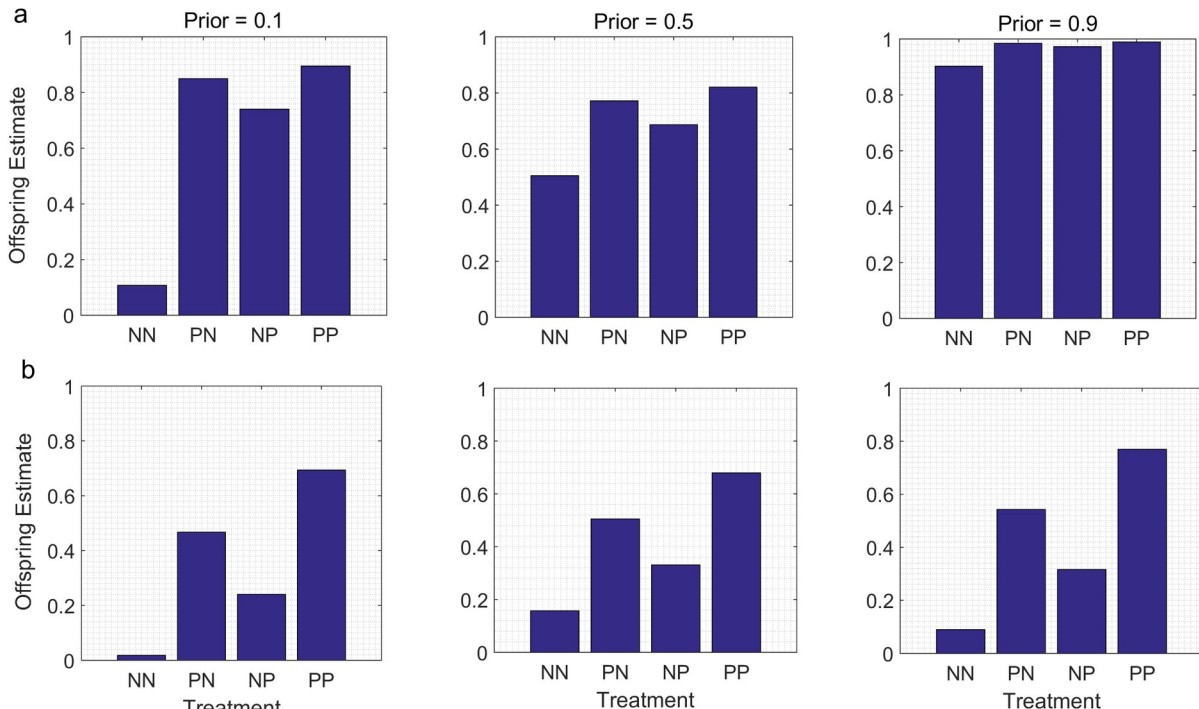

**Fig 3. Duration of exposure to the same cues is longer for parents than for offspring.** In the P treatment groups, parents are exposed to the presence of cues from predators for time four intervals, but offspring are exposed to cues from predators for only two of the four time intervals (see text). Other variables are the same as in Fig 2.

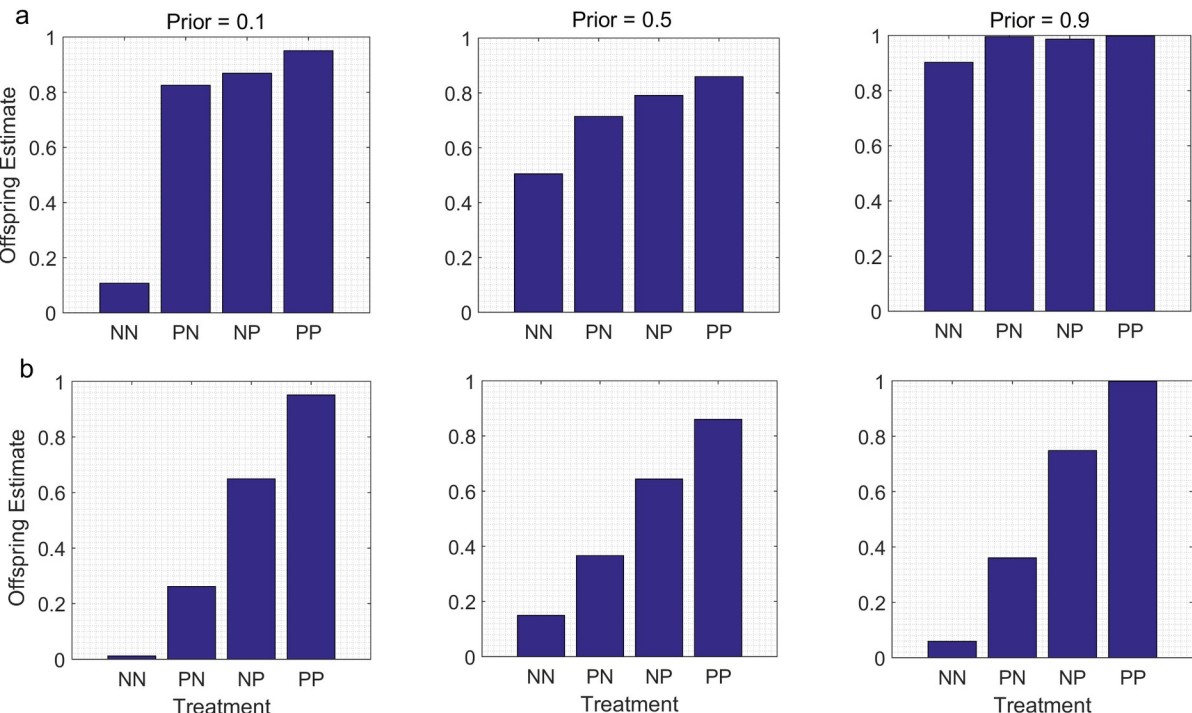

**Fig 4. Information from parents is devalued or degraded before being passed to the offspring.** We assume that parents and offspring in the P groups are exposed to the same cue for the same period of time, with a likelihood function indicated by $\alpha = 8$, $\beta = 1$. However, the signal provided by the parent to the offspring as a result of the parent's exposure to the cue is less reliable; it is modelled by a likelihood indicated by $\alpha = 3.5$, $\beta = 0.44$. a. N- model: information provided by conditions in the N treatment is much less reliable than the information provided by conditions in the P treatment (likelihood for conditions in the N treatment modelled by $\alpha = 1$, $\beta = 1$). b. N* model: conditions in the N treatment provide information that is as reliable as conditions in the P treatment (likelihood for N modelled by $\alpha = 1$, $\beta = 8$). However, the signal that parents provide to their offspring as a result of exposure to conditions in the N treatment is less reliable, with a likelihood modelled by $\alpha = 0.44$, $\beta = 3.5$.

patterns were otherwise similar to those in the baseline models. A jump-up pattern was detectable in the N- models when the parental Prior indicated a different value of the state of the environment than the cues in the P treatment, but not when the parental Prior indicated a value similar to that indicated by the cues in the P treatment (Fig 4A). In contrast, similar step-up patterns were detectable for all parental Priors in the N* models (Fig 4B). However, for comparable sets of parameter values, the differences between the offspring estimates for the PN and NP groups were much less pronounced in the N- models (Fig 4A) than in the N* models (Fig 4B).

## Discussion

The current study suggests that empiricists should not be surprised to observe considerable variation in the results of experimental studies of trans and within generational plasticity (TWP) in response to informative cues. Our results show that the offspring information-states that are assumed to provide the foundation for variation among the treatment groups in offspring trait values in experimental studies of TWP are expected to vary as a function of many factors, including 1) parental Prior distributions, 2) the relative reliability of the information provided in the different treatments, 3) differences in the duration of exposure to the same cues for parents and offspring, 4) the extent to which information based on cue-exposure for parents is devalued, relative to the information based on cue-exposure for offspring, and 5) the

extent to which information passed from parents to offspring is degraded before it is passed along to the offspring. We illustrate these findings by modelling fully factorial experiments in which parents, offspring, both, or neither are exposed to cues from predators (in P treatments) or are not exposed to those cues (N treatments).

Our results confirm previous findings that interactions between parental Priors and relative cue reliability can have a major impact on the patterns of offspring information-states in empirical studies of TWP [30]. We analyzed two extreme situations for differences in the reliabilities for the two treatments: N- models, in which the information provided by conditions in the N treatment is much less reliable than the information provided by conditions in the P treatment, and N* models, in which the information provided by the conditions in the P and the N treatments is equally reliable. A possible example of the former might be an experiment in which the cues from the predator in the P treatment consist of a single, near-escape from a predator over a period of several months. In this case, conditions in the P treatment might provide a reliable indication that the predator in question lives at the current locality, whereas the conditions in the N treatment might provide a less reliable indication that it does not. By way of analogy, being the victim of a robbery once over the course of a year might indicate with a high degree of reliability that thieves are active in your neighborhood, but if such incidents rarely occur on a per-capita basis, not being robbed over the course of a year may not indicate with the same level of reliability that they are not. An example of a situation in which the conditions in the P and the N treatments might provide equally reliable information is when the subjects in the P treatment are exposed for an extended period to a high concentration of kairomones from a predator and the subjects in the N treatment are maintained in the absence of kairomones for the same period of time. In this situation, investigators typically assume that the conditions in the P treatment reliably indicate that predator density is high, whereas the conditions in the N treatment indicate with a comparable level of reliability that predator density is low, since in that treatment the concentration of kairomones never exceeded the preys' threshold for detection (see [30] for additional discussion).

The current results also confirm previous findings that the patterns of offspring estimates were highly dependent on parental Priors when the information provided by the different treatments differed in its reliability (as in the N- models), but not when the information provided by the different treatments was equally reliable (as in the N* models) [30]. These results imply that if investigators use an experimental protocol in which the information provided by the different treatments differs in its reliability, the results of their experiments might vary as a function of the population-of-origin or the genotypes of their subjects. This follows from the assumption that subjects from different populations might have different parental Priors based on information from their ancestors, e.g., via genes, inherited epigenetic factors or grandparental effects (e.g. [16, 20, 24]). In addition, if the parents for an experimental study of TWP were collected from the wild, then variation among populations in parental experiences earlier in life could also contribute to variation among populations in parental Priors. Along the same lines, different genotypes from the same population might begin with different prior estimates of the value of a given state of the environment (e.g. [26]), so that the results of studies of TWP using clonal organisms might vary, depending on the clone that was selected for the study.

Indeed, if investigators used an experimental protocol in which the conditions in the different treatments greatly differed with respect to the reliability of their information, the choice of population or genotype for a study might determine whether one could even detect plasticity. For instance, in all of the N- models analyzed in this article, a 'jump-up' pattern (NN < PN, NP < PP) for offspring estimates was observed when the value of the state of the environment indicated by the parental Prior strongly contradicted the value of the state indicated by the cues in the P treatment. However, when the value of the state indicated by the parental Prior

was similar to the value indicated by the cues in the P treatment, the offspring estimates were virtually identical in all four treatment groups (NN≈ PN≈ NP≈ PP). In the latter situation, one would not expect to detect WGP, TGP or TWP.

In contrast, consider a situation in which an investigator used an experimental protocol in which the cues in the different treatments were all equally reliable (as in the N* models). In this case, our results suggest that the choice of a population or a genotype for the experiment might not matter. As we show here for all of the models, in this situation we expect to observe step-up patterns (NN < PN,NP < PP), regardless of the parental Prior.

As intuition suggests, the main effect of assuming that cue duration differed for parents and offspring or that the information provided by parents was either devalued or degraded was to generate different values of the offspring estimate for the PN and the NP treatments; in the absence of these assumptions, the offspring estimates were the same for the PN and NP treatments. However, for the same parental Prior and for the same cues in the P treatment, the effects of differences in cue duration or the devaluation or degradation of information from parents were much more pronounced in the N* models than in the N- models. This suggests that in general, it would be easier to detect the effects of cue duration or of the parental devaluation/degradation of information on offspring estimates or trait values in TWP studies if one used an experimental protocol in which the information provided by both treatments was similarly reliable than if one used a protocol in which the information provided by one treatment was much more reliable than the information provided by the other.

If we just focus on situations in which the information provided by the conditions in both treatments is equally reliable (the N* models), the results indicate, as intuition would suggest, that the effects of those exposures on offspring estimates of the value of the state would be stronger for the generation that was exposed to the cues for the longer period of time. This result supports earlier suggestions that parental experiences might have a stronger impact on offspring trait values than offspring experiences, because the cumulative experiences of parents over the course of their lifetimes provide more reliable information than is available to offspring based on their own experiences during the limited period from birth until their phenotypes are measured [38, 39]. As a practical matter, these results indicate that variation among experimental studies of TWP in the relative duration of exposure to the same cues for parents and offspring would, all else being equal, be expected to contribute to variation in their results.

The models of information-updating presented here may also provide a useful point of departure for investigating other aspects of the timing of cue-exposure that might contribute to the variation results observed in empirical studies of TWP. For instance, the models analyzed herein ignore sensitive periods, that is, situations in which exposure to the same cues have different effects on trait values, depending on the age of onset of the cue-exposures [25, 40–42]. Traditionally the literature on sensitive periods has focused on the effects of an individual's own experiences early in life on its trait values later in life (i.e. WGP); thus far, Bayesian models of sensitive periods have also focused on this situation (see review in [25]). Recently, however, empiricists have begun to consider the possibility of sensitive periods for TGP, i.e. situations in which exposure to cues for parents have different effects on the trait values of their offspring, depending on the age at which those exposures occurred in the parents, or the age at which the offspring received the information from their parents (e.g. [31, 33, 43]). However, at this point it is unclear how sensitive periods of parents, offspring or both might interact to affect offspring estimates of conditions in the external environment.

As a general rule, we would expect the effects of sensitive periods on offspring information-states to depend on the protocol used in a given experiment. Personal exposure to cues in the external environment can't begin to affect the information-states of offspring until the offspring are capable of detecting those cues (e.g. as embryos, newborns, or hatchlings). An

additional complication is that in order to demonstrate that cue-exposure for parents affects the trait values of their offspring, exposure of parents to cues from the external environment must end before their offspring become capable of detecting the cues on their own. Hence, if both parents and offspring were continuously or repeatedly exposed to the cues of interest from the embryo stage until just prior to offspring production, both generations would be exposed to those cues throughout their sensitive periods, no matter if or when they occurred in either generation. In that case, the predicted patterns of offspring estimates would be the same as those described in the current article, in the absence of sensitive periods. However, if parents or offspring had sensitive periods for the effects on cue-exposures on offspring trait values, if these periods occurred at different ages in the two generations, and if exposure to the cues was limited to restricted periods in both generations, then the effects of sensitive periods on offspring estimates are unclear. This would be a worthwhile topic for additional theoretical and empirical study.

Our results also confirm the intuitive notion that when information from parents is devalued relative to information from offspring, exposure to cues for parents would have a weaker effect on offspring information-states than exposure to the same cues for their offspring. Theory predicts that the devaluation of information from parents should occur as an evolved, adaptive response to reduced levels of autocorrelation between parental environments and offspring environments (see Introduction). In other words, the devaluation of information from parents is expected when, under natural conditions, the value of a state of the environment is likely to change between the time that parents are exposed to cues and the time that their offspring are exposed to the same cues. For instance, if we assume that the true value of the state gradually changes over time, the devaluation of information from parents would be inversely related to the amount of time that elapsed between the time that parents were exposed to cues and the time that their offspring were exposed to the same cues. In turn, this implies that for subjects from the same population, the devaluation of information from parents would be less pronounced in an experiment in which parents were exposed to the cues of interest just before their offspring were conceived and their offspring were exposed to the same cues soon after hatching than in an experiment in which parents and offspring were both exposed to the cues soon after hatching.

Of course, in natural populations, the true value of a state of the environment need not change gradually over time, but may instead occur at specific times, lifestages or life-history landmarks. For instance, in a species in which the value of a state of the environment (e.g., predator density) does not change over the distances typically traveled by natal dispersers, information based on a parent's experiences in its natal habitat would also be relevant to the conditions in its offspring's natal habitat. In contrast, in a species in which the value of a state varies over typical dispersal distances, and in which parents disperse prior to reproducing, information based on a parent's experiences in its natal habitat would be less useful for estimating the conditions in their offspring's natal habitat. In other words, theory suggests that we should expect the devaluation of information from parents to differ among populations or closely-related species, depending on the extent to which the environmental factors of interest vary spatially over the distances relevant to natal dispersal, or vary temporally within and across generations [7, 11, 14, 32]. Hence, one could potentially test predictions about the devaluation of information from parents using comparative data from field studies indicating whether, how, and when particular environment conditions of interest are spatially and temporally correlated for parents and their offspring. Unfortunately, at present these data are still sparse [44–46].

In addition, our results also indicate that if information from parents is degraded before it is passed along to their offspring, exposure to cues for parents would have a weaker effect on

offspring estimates than exposure to the same cues in offspring. The degradation of information from parents is assumed to occur as a non-adaptive consequence of the proximal mechanisms by which parents transfer information to their offspring. That is, the degradation of information from parents to their offspring is assumed to be a result of the 'noise' introduced into the flow of information from one generation to the next, based on the series of processes that intervene between the time that mothers and/or fathers are exposed to the cues and the time that their offspring receive signals from their parents based on the parent's experiences earlier in life [33].

Because we assume that the degradation of information occurs as a result of unavoidable inefficiencies in the mechanisms that are responsible for the transfer of information across generations, we would expect the degradation of information from parents to be phylogenetically conservative. That is, because populations and species are expected to have comparable patterns of parental care and rely on comparable proximate mechanisms to transfer information from parents to offspring, we would not the degradation of information from parents to vary much across populations or closely-related species. Instead, testing hypotheses about the effects of the degradation of information from parents to offspring will require detailed information about the proximal mechanisms which mediate the flow of information from parents to offspring in a given taxon. At present, little is known about this topic [33]; this would be another fruitful subject for future research.

Under limited conditions, models which predict offspring information-states may also provide insights into the phenotypes we might expect to observe in empirical studies of TWP. One important condition is that the experiences which induce changes in the development of phenotypic traits in organisms (here, called 'inductive experiences') only provide information, as opposed to also having direct, long-lasting effects on the somatic states of parents, offspring, or both. Detailed discussion of the difference between inductive experiences which affect development because they provide information versus inductive experiences which affect development because they have persistent effects on somatic states can be found in [6, 9, 47, 48]. An example of an inductive experience with a direct effect on development is rearing temperature: we would expect the temperatures to which organisms are exposed during the juvenile period to have immediate and long-lasting effects on their growth and development, regardless of whether or not those temperatures also provide them with information about the temperatures they are likely to encounter later in life. An example of an information-only inductive experience is a chemical produced by a predator: this stimulus only affects growth and development because it triggers a particular response in the nervous system, presumably because it provides information about the predators which produce this chemical. Examples of other inductive experiences which might provide information, but which also have direct effects on a developing organism's somatic state, include food deprivation in animals or shade in plants. Examples of other information-only inductive stimuli include stimuli produced by conspecifics, hosts, or habitats. As Nettle and Bateson (2015) point out, if an inductive experience is information-only, one can imagine a single loss-of-function mutation that abolishes an individual's ability to detect that cue, but which leaves the developing individual otherwise unaffected. In contrast, if an inductive experience has a direct and lasting impact on an individual's somatic state, we would expect that experience to affect its development even if the individual was unable to sense that it had that experience. Although it is possible to make general predictions about how the development of phenotypic traits might change in response to exposure to information-only cues over the course of development (e.g. [18, 21]), this is much more difficult when inductive experiences not only provide information, but also have direct, lasting effects on an individual's metabolism, growth, or other aspects of its somatic state (but see [17]).

A second important condition for assuming relationships between offspring estimates and offspring phenotypes is evidence that the phenotypic trait is advantageous when the offspring is in the environment indicated by the cues in the experiment. For example, in empirical studies of WGP and TGP in response to cues from predators, investigators often focus on inducible defenses, e.g., behavioral or morphological traits that have been shown to increase survivorship when individuals are in the presence of predators. In such cases, it is reasonable to assume that those traits would be more strongly expressed if a subject's estimate of the value of an environmental state that contributes to predation risk (e.g., predator density) was high than if it was low. In contrast, analyses of information-states are much less useful for predicting the developmental plasticity of traits for which the expected adaptive response to different information-states is unknown or uncertain.

However, if a cue is information-only, if the adaptive phenotypic response to that cue is clear, and if one has a reasonable idea about the relative reliability of the information provided by the conditions in the different treatments, then these models can provide a useful benchmark against which to compare the patterns of offspring trait values expressed in empirical studies of TWP. For instance, if parents and offspring are both exposed to the same cues from predators from birth to maturity, if one measures inducible traits known to improve survivorship in the presence of that predator, and if it is reasonable to assume that the cues in the P treatment and the cues in the N treatment are equally reliable, then the models in the current paper predict a 'step-up' pattern for offspring trait values, in which the values for an antipredator trait for the PN group are either the same or lower than the values for the NP group (see Figs 1B and 4B). In addition, if information from the field indicates that the environmental state of interest might change between the parental and offspring generations, one would expect the trait values for the NP group to be higher than the trait values for the PN group.

An example of an experiment for which these conditions appear to be satisfied is a classic study of TWP in which *Daphnia cucullata* parents and offspring were exposed to kairomones from a predator (*Chaoborus flavicans*) from birth to first reproduction, and then relative helmet length was measured at the age of first reproduction in the offspring [2]. Other experiments have shown that kairomones from *C. flavicans* induce the development of larger helmets in *D. cucullata* [49], and that larger helmets protect juvenile and adult *D. cucullata* from this predator [50]. In addition, field studies of *C. flavicans* indicate that the risk it poses to *Daphnia* spp. varies within seasons across the temporal scales that might encourage the devaluation of information from parents [51]. Finally, continuous exposure of kairomones from a predator from birth to maturity in a P treatment is typically assumed to provide reasonably reliable information about the density of that predator, while the absence of kairomones from birth to maturity in an N treatment is assumed to provide comparably reliable information indicating that the density of that predator is low (see [30]). As predicted by the models described herein for this set of conditions, [2] reported a step-up pattern (NN < PN < NP < PP), in which the trait values for all four groups were significantly different from one another.

Finally, the current study shows how an appreciation of the ways that information from different sources is expected to combine within and across generations reveals a number of questions that investigators might want to consider when they are planning or interpreting the results of empirical studies of TWP in response to inductive cues. These are summarized as follows:

1. Are the inductive cues information-only, or could the inductive experiences also have direct, lasting effects on the somatic states of either the parent or their offspring? Insights

from models of development based on information-updating are currently most useful for predicting trait values when cues are information-only.

2. Are the conditions in the different treatments likely to provide equally reliable information about a state of the environment, or are the conditions in one treatment likely to provide much more reliable information than those in the other treatment? As was shown here and in [30], we expect the patterns of offspring estimates in TWP studies to dramatically differ in these two situations.

3. Are parents and offspring exposed to the same cues for the same period of time? As is shown in the current article, all else being equal, differences in the duration of cue-exposure for parents and offspring are expected to affect the patterns of offspring estimates, especially if the cues in the different treatments are equally reliable.

4. Is there strong existing support for the assumption that a particular response in a particular trait to a particular cue is likely to be adaptive? In such cases, it is more likely that differences among the treatment groups in offspring information-states at the end of the experiment will be related to differences among those groups in the trait values expressed by the offspring at the end of the experiment.

Finally, our results show that it may not be necessary to invoke assumptions about developmental constraints, costs of sampling, the fitness consequences of trait values for offspring or other factors to account for at least some of considerable variation in results that empiricists have observed in their studies of TWP (e.g. [31]). Instead, we show that models based on first principles of information-updating generate complicated, often non-intuitive, patterns of the information-states that are assumed to provide the foundation for much of the variation in offspring phenotypes observed in experimental studies of TWP in response to cues. Considerable variation in the patterns of offspring information-states can be produced by interactions among factors which are likely to vary among experiments, including parental priors, differences in the reliability of the cues in the different treatments, differences in the duration of exposure to the same cues for parents and offspring, and the extent to which information from parents is devalued or degraded before it is passed along to the offspring. We suggest that empiricists studying TWP consider how basic components of information-updating might contribute to their results, before assuming that other factors which affect development are required to explain them.

## Supporting information

**S1 Appendix. List of terms.**
(DOCX)

**S2 Appendix. Estimating posterior probabilities using Bayesian estimation.**
(DOCX)

**S3 Appendix. Specifying the cumulative likelihood functions used to describe the information provided by the parent's and offspring's experiences in studies of TWP.**
(DOCX)

**S4 Appendix. Pseudo-code for the Matlab program used to analyze TWP.**
(DOCX)

**S5 Appendix. Matlab code for the program used to compute and plot results for Bayesian analyses of TWP.**
(DOCX)

**S1 Fig. Parental prior distributions.**
(DOCX)

**S2 Fig. Cumulative likelihood functions.**
(DOCX)

## Acknowledgments

We thank Yifeng Xu for help with programing, and Marc Mangel and Juliette Tariel for their comments and suggestions on ways to improve the manuscript.

## Author Contributions

**Conceptualization:** Judy A. Stamps, Alison M. Bell.

**Formal analysis:** Judy A. Stamps.

**Investigation:** Judy A. Stamps.

**Methodology:** Judy A. Stamps.

**Project administration:** Judy A. Stamps.

**Writing – original draft:** Judy A. Stamps, Alison M. Bell.

**Writing – review & editing:** Judy A. Stamps, Alison M. Bell.

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
