## [Decision Letter · Decision Letter 0]

15 May 2021

PONE-D-21-10936

How basic components of information-updating interact to encourage variation in the results of empirical studies of within and transgenerational plasticity

PLOS ONE

Dear Dr. Stamps,

Thank you for submitting your manuscript to PLOS ONE. After careful consideration, we feel that it has merit but does not fully meet PLOS ONE’s publication criteria as it currently stands. Therefore, we invite you to submit a revised version of the manuscript that addresses the points raised during the review process.

Your manuscript has been seen by two referees, both of which are highlighting that your contribution is an important and valuable extension of previous work on intra- and trans-generational plasticity. Nonetheless, the manuscript needs some work regarding methodological details, especially description of the model, and clarification of textual aspects. It would be good to briefly describe the model this work builds on, to provide a figure how the Bayesian updating works and add other details of the model in a supplementary file.

We look forward to receiving your revised manuscript.

Kind regards,

Peter Schausberger

Academic Editor

PLOS ONE

Journal Requirements:

[This material is partially based upon work supported by the National Science Foundation under Grant No. IOS 1121980 and the National Institutes of Health under award number 2R01GM082937-06A1.]

 [The authors received no specific funding for this work. ]

Reviewers' comments:

Reviewer's Responses to Questions

**Comments to the Author**

1. Is the manuscript technically sound, and do the data support the conclusions?

Reviewer #1: Yes

Reviewer #2: Yes

2. Has the statistical analysis been performed appropriately and rigorously? 

Reviewer #1: N/A

Reviewer #2: Yes

3. Have the authors made all data underlying the findings in their manuscript fully available?

Reviewer #1: Yes

Reviewer #2: No

4. Is the manuscript presented in an intelligible fashion and written in standard English?

Reviewer #1: Yes

Reviewer #2: Yes

5. Review Comments to the Author

Reviewer #1: As you will see from my review, I think that you have done a masterful job explaining the modeling results but think that the paper will be much stronger if you add a SM with some more details of how the model works.

Reviewer #2: In this study, the authors use Bayesian models to investigate whether basic components of information-updating explain the wide diversity of patterns we see in transgenerational plasticity studies. This study is the follow-up of another study published in Oecologia in 2020. The authors went further in their analysis by adding two parameters (difference in duration of cue exposure and in cue importance between parents and offspring) which are susceptible to interact with the previous parameters they described, and which often vary in experimental studies on transgenerational plasticity. They found that these two new parameters are indeed important for information-updating as offspring estimates of its environment depends on interactions between the new and the previous parameters investigated. I do not have many suggestions to make because the manuscript is really good and a highly valid contribution to the scientific record. I think that is because 1) this paper is the follow-up of another paper which has already been published, so from which the authors already get a lot of comments, and 2) both authors are (from my point of view) really good writers and scientists. The only weakness I see is that the authors did not mention the software they used for the analysis, makes it impossible to repeat their results.

It is my first review for PLOS ONE, I do not know what the standards are to make the report, so I 'm just going to go through all the publication criteria step by step.

1. The study presents the results of primary scientific research.

Yes.

2. Results reported have not been published elsewhere.

They have not.

3. Experiments, statistics, and other analyses are performed to a high technical standard and are described in sufficient detail.

Yes, except that:

- you do not mention what software you use for the simulations. I don’t think (but I might be wrong) that you do not calculate the mean and variance of posterior distributions by hand, so you must have use a software.

- If a reader wants to fully understand your methods, he/she has to look for the additional information of your previous papers (quote [25]). It is not very convenient. Can you provide an additional information also for this current paper where you just copy-paste the information that you put in your last paper? Or provide at least a direct link to the additional information of your previous paper?

4. Conclusions are presented in an appropriate fashion and are supported by the data.

Yes, what the authors claim is supported by the data and presented in an appropriate fashion.

Just a minor comment, I find that the discussion lacks references. I know that you are probably going to repeat the references you put in the introduction, but a reader that just want to read the discussion needs to have the references in the discussion too. For example, I find that the sentences L409-412, L503-507 and L532 need references.

5. The article is presented in an intelligible fashion and is written in standard English.

Yes. I find both authors good writers in other papers that I could read, and this one is not an exception. The paper is very pedagogic and written with a high-standard English. It is very nice because each time a sentence is a little bit complicated to understand, there is an example after, so we get all what the authors want to say.

6. The research meets all applicable standards for the ethics of experimentation and research integrity.

Yes.

7. The article adheres to appropriate reporting guidelines and community standards for data availability.

The data and analysis are appropriately reported. However, the authors say that the data are available, but they do not say where.

Minor comments

Introduction

L14-18: the sentence is clear, but I suggest cutting this very long sentence

L45: Is there any experimental evidence that individuals indeed use Bayesian updating to estimate their environment?

L58-69: this paragraph is less clear compared to the rest of the introduction. I suggest a little reorganization:

1) TWP studies focuses on phenotypic values (L58-59)

2) Problem: information-based models tell us mostly about information-states and to have access to phenotypic values on information-based models, you need to include much more factors and assumptions. And if you include many assumptions, you can’t have access to how much information-updating contributes to the phenotypic values (L65-69)

3) the solution is to focus on information-states (L70-71). Not a problem because “when WGP, TGP or TWP occur in response to cues that provide information about the external environment, variation among individuals in their information-states provides the foundation for variation in the trait values observed in experimental studies of these phenomena” (L59-62). Focusing on information-state has already given successful results in other fields (L71-82).

4) What have been done on TWP using information-updating Bayesian models (L83-87)

5) However, currently, nothing known about key factors that impact information-updating (L62-64) and which vary across experimental studies.

L121-126: A small comment but I do not understand why you describe these results so thoroughly. This paragraph is focused on the devaluation of parents’ information compared to offspring’s information, not on the importance of temporal autocorrelation.

L134: it is confusing for a reader to see twice “second” (L113 and L134). I understand after reading again that you want to say that there is two reasons why the assumption L113-155 is not valid. I suggest to say “the first reason is” at the L115 and the “second reason is” at the L134 or to put “third” at the L134 as this sentence refer to the third variable (L91).

L127-133: I would not start a new paragraph, I would do only one paragraph (L113-133), as the ideas of this paragraph (L127-L133) are connected to the previous one.

L145: little bug? “and/or”

Methods

It is just a suggestion, but I think it will be clearer to see the prior distribution on multiples graphs that you could put in supporting information. I saw after that you did it for your previous paper, which I find very neat.

L228: little bug, I have a comma that is highlighted in red as when the modifications are tracked in Word.

L243: bug, two times “instance”

Discussion

L445: bug, a space is highlighted in red.

L524-525: this sentence is confusing because with “i.e.”, it looks like you say that species with similar offspring development and parental care share the same constraints of cue degradation. But I think you want to say that closely related species or populations share the same mechanisms to pass on cue, precisely because they are closely related. I would remove “i.e. similar for populations and species with comparable patterns of offspring development and parental care.”

But maybe I did not understand the sentence.

L539: I am sure you know this, but as stated, it sounds like you think that information-only inductive experiences do not affect the somatic state of parents. Information-only inductive experiences also generate effect on the parental state (e.g. trade-offs between production of defences and other functions, change in life-history strategy), which can have an effect on offspring trait value in addition to the transmission of information from parents to offspring. I would be nice to have a mention of that, otherwise it seems too simplistic (from my point of view).

L533-534: I think you mention in the introduction that it is difficult to link information-state to phenotypic values because of costs of sampling, developmental constraints, or the fitness consequences of expressing different trait values, but this does not appear in the discussion. Even though the point of your paper is to show that information-updating is sufficient to explain the diversity of TWP patterns, I think a few sentences on this topic would be nice to show that it is complicated because these parameters can influence how differences in offspring estimates translate into differences in phenotypic values.

L592-608: just a useless comment but I really like the questions at the end of the discussion dedicated to investigators. I think it is easy to read, instructive and pedagogic.

L596: it would be nice to have some papers or ideas on how estimate the difference in reliability between treatments.

6. PLOS authors have the option to publish the peer review history of their article (what does this mean?). If published, this will include your full peer review and any attached files.

Reviewer #1: **Yes: **Marc Mangel

Reviewer #2: **Yes: **Tariel Juliette

---

## [Author Response · Author response to Decision Letter 0]

3 Jun 2021

Please the attached file (Response to reviewers)...because it provides this material in a format which may be easier for reviewers to follow...

Editor

Your manuscript has been seen by two referees, both of which are highlighting that your contribution is an important and valuable extension of previous work on intra- and trans-generational plasticity. Nonetheless, the manuscript needs some work regarding methodological details, especially description of the model, and clarification of textual aspects. It would be good to briefly describe the model this work builds on, to provide a figure how the Bayesian updating works and add other details of the model in a supplementary file.

Done. We have extensively rewritten the text, and now include four new Appendices (S2-S5) and two new figures (Figure S1 and Figure S2) to explain how the model works

Changes to the list of references:

In response to the requests of the two reviewers, we added several references to the paper: #17, 27,28 and 29. 

In response to a suggestion from one of the reviewers, we have removed two references to the paper:

Lind et al. 2020, and Dey et al. 2016. 

Reviewer #1: Mangel review

1. lines 19, 35-36: I suggest that you write “trans and within generational plasticity” so that the

acronym TWP makes sense.

Done (throughout the paper)

2. There are people who will get bent out of shape when you say that Bayesian methods

are the best, so you could better write this as “..Bayesian methods are a consistent and natural

way to combine...”

Done

3. line 63: will most readers know what naive priors are? It would be good to have a note

indicating that this is described in the SM glossary (to which it needs to be added).

Done. We now define “naïve prior” in the S1 Appendix 

4. line 82: Exactly what empirical results are of interest here? If they are in any of your

papers cited, then please indicate which ones. If not, think about adding a table concerning the

empirical work that you are wanting to explain

We have rewritten this section of the text to indicate that we are referring here to situations in which developmental plasticity in response to the same experiences/cues varies among individuals, populations or species, and cite several review articles to illustrate this point. 

5. lines 90-92 and throughout: As far as I can tell, this is the first time that the words devalued

and degraded are used.

We have rewritten this section to describe in more detail what we mean (see lines 126-158) 

6. lines 113, 127, 134: These begin “Second”, “In addition”, “Second” and I lost the flow.

Please fix that.

Done. This section has been rewritten

7. lines 149-153: It took eight pages to get to an explanation of what you are doing, but it will 

be hard for readers to stick with it. I suggest that you move this up around line 103 and then

let the following writing show why this is worth doing.

Done. This introductory material has been moved up to lines 103-110.

8. lines 156-173 and throughout. Why does prior have an upper case P? You also need to let the

reader know what random variable this prior is characterizing. Please think about something

other than saying “state of the environment” but rather give an example or two

We now give examples of states of the environment in line 165, and explain why we capitalize the parental Prior in lines 177-180.

9. lines 191-194: How does gamete production get into here? I thought that the point was to

focus on information states rather than detailed development

We have omitted this sentence, to avoid confusion. But although this article does not focus on ‘details’ of development, it is all about the changes in information-state over ontogeny that are assumed to provide the basis for changes in phenotypic development in response to informative cues. See below

10. lines 199-206: I cannot figure out what is going on here. In lines 203-204 do you mean

probability of exposure to a predator or not?

11. lines 212-216: This is a nice overview of what you did and should come sooner.

This section of the methods has been extensively rewritten to clarify what is going on… 

12. lines 238-239. Now you are talking about variance of the likelihood, by which I presume you

mean the entire beta density. It would be very good for the reader to thus know how the mean

and variance of a beta density depends on its parameters, without having to go back to Stamps

and Krishnan (2014a). All of these issues can be fixed if you have a SM figure that shows the

three beta densities that you use.

This was a good suggestion. We now include a new figure (S1) which shows the beta distributions used to generate the shapes of all of the cumulative likelihood functions used in the current study. 

13. line 299 following. It is not clear to me if these are captions for figures or discussion of the

figures. But most importantly – what is unexpected in these results?

These were the captions for the figures, placed as requested by the journal. We discuss aspects of the results that were ‘unexpected’ in the Results and Discussion sections

14. lines 448-452. You have already noted that one of the properties of Bayesian updating is

that unless information is forgotten, the posterior usually is tighter (smaller variance) than the

prior, so it is almost an a priori prediction that the lifetime experience of the parent provided

to the offspring will have a stronger impact than early lifetime experiences of the o↵spring. Am

I missing something?

We have rewritten this section to improve clarity. Kuzawa suggested that if human parents were exposed to cues for many years but their offspring were exposed to the same cues for a few months before the traits were measured in the offspring, parents would have a better estimate of conditions in the external environment than would their offspring. In Bayesian models this would occur because the variance of the prior declines as a function of the amount of time that subjects are exposed to the same cues. However, Kuzawa made his suggestion in the absence of any formal models of information-updating.

15. lines 453-464. You began the paper saying that it you were focussing on information without

the developmental state, so I suggest you let readers fill in this paragraph in their own minds.

This also makes a smoother transition to the paragraph beginning on line 465

We obviously needed to clarify that this paper IS about development…but instead of focusing on the development of phenotypes, we are interested in how exposure to cues affects the development of differences among individuals in their information-states, because these differences in information-state are assumed to provide the foundation for the differences in the phenotypic traits that develop and are subsequently expressed by those individuals. 

We have rewritten the introduction to stress that we are interested in development, developmental plasticity, changes in information-state over ontogeny, etc. etc. in this paper…

16. lines 496-510. I am not sure that you need this paragraph for the current manuscript, since

you do not treat this question. If you decide to keep it, fine, but in line 510 the word data is

plural so the sentence should read “Unfortunately, at present these data are still sparse [42-44].”

We retained this section because it is relevant to the question of how one would study the devaluation versus the degradation of information from parents. As is described later, one might observe variation among populations or closely-related species in the devaluation of information from parents, but we would not expect comparable variation at lower taxonomic levels for the degradation of information from parents.

17. lines 511-532: Much of this is about degradation of information from the parents, but I still

do not understand how that is treated in the model or what you mean by ‘noise’ (or why it is

in quotation marks).

We have rewritten the section in the introduction to clarify what we mean by the devaluation or degradation of information from the parents, and rewritten the section in the methods to clarify how we modelled these. 

18. lines 533-540: This is the first time that the word inductive is used in the paper and it is

not in your little glossary. I cannot figure out what this paragraph is about since I do not know

what inductive cues or experiences are

As is indicated now in line 593, “ inductive experiences” are simply experiences which induce changes in the development of phenotypic traits. Inductive experience is now added to the list of terms in the S1 Appendix

19. lines 552-587: This is all about development and I wonder if you need it. If you are setting

up future work in which this paper is linked to models of development, okay. But otherwise

perhaps not.

As was noted above (see 15), this paper is one of a series of Bayesian models of development… the point of all of these models is to describe how changes in an organism’s information-state over ontogeny might affect the development of phenotypic traits in that organism. In some cases (e.g. the Frankenhuis series of models), the models include a long list of assumptions aside from those required for information-updating…these added assumptions are required for models to predict phenotypic traits, as opposed to information-states. 

The point we are trying to make in this section is that a direct mapping of information-state to phenotype is only likely to occur when certain conditions are met…e.g. that the inductive cues be information-only (e.g. stimuli from predators, as opposed to food intake), and that researchers have a good idea before they start the experiment about the expected relationship between information-state and phenotype (e.g. they might reasonably expect a positive, saturating relationship between an individual’s estimate of predator density and the level of antipredator traits that develop in that individual). 

20. line 683. This citation needs a date. 

We are confused…the date (2020) for this citation was in the same location as the date for the other references:

26. Tariel J, Plenet S, Luquet E. Transgenerational plasticity in the context of predator-prey

684 interactions. Front Ecol Evol. 2020;8. doi: doi: 10.3389/fevo.2020.548660.

21. line 735, 739. Species name in italics. 

We are not sure whether PlosOne requires this….we can add these if it is not done automatically when the references are entered by the journal

22. Figures: In all the figures, the prior is not 0.1, 0.5, or 0.9. Rather the mean of the prior

takes those values.

Yes, we have indicated this in the text and in the figure legends. But since the variance of the prior is the same for all of the models in this article (0.04), we wanted to stress that the variable which differed in our analyses was the MEAN of the parental Prior. 

23. Why is the estimate only the mean of the prior or posterior distribution? The estimate is

the entire distribution, you are choosing to use the mean – which is fine since you do not have

any development in here. When development is a nonlinear function of the state, then we expect

the entire distribution to play a role in development, not just its mean

This is true in general. However, as we now note in lines 234-238, the variance of the posterior for the offspring at the end of the experiments was similarly low across most of the treatments for most of the models. This was because most of the cues to which the subjects were exposed over the two generations were moderately to highly reliable. It would be interesting to consider how the development of phenotypes would respond to a situation in which offspring ended up with posterior distributions with similar means but very different variances, but that situation did not arise in the current study.

 24. Although the code is available, it is often very difficult to read other people’s code, especially if

one is trying to grasp the logic of the model rather than the details. To help readers who want to replicate your work or build on it, I suggest that you provide either or both of

• A flowchart showing how the model works.

• Pseudo-code, which can refer to commonly used distributions such as the beta (e.g. a

statistics book or an R library) but which gives equations for the unusual ones in your

work.

We have provided a number of new supplemental files to describe how the model works. These include S2 Appendix (Estimating posterior probabilities using Bayesian estimation), S3 Appendix (Specifying the cumulative likelihood functions used to describe the information provided by the parents’ and offspring’s experiences in studies of TWP), S4 Appendix (the pseudo-code for the Matlab program we used to compute and plot the results of our Bayesian analyses of TWP) and S5 Appendix (the Matlab program we used to compute and plot the results of our Bayesian analyses of TWP). 

In addition, we have extensively rewritten the first portion of the methods section and have expanded the list of terms (S1 Appendix) to explain in more detail the design and assumptions of the models. 

Reviewer #2: Tariel Review

1. The only weakness I see is that the authors did not mention the software they used for the analysis, makes it impossible to repeat their results.

Both reviewers asked for more information about the models, so we have expanded the explanations of the models in the Methods section, and have added a number of supporting files to the paper, including not only the program that was used to analyze and report the results, but also pseud-code to explain how the program works (see Mangel review, above).

2. Conclusions are presented in an appropriate fashion and are supported by the data.

Yes, what the authors claim is supported by the data and presented in an appropriate fashion.

Just a minor comment, I find that the discussion lacks references. I know that you are probably going to repeat the references you put in the introduction, but a reader that just want to read the discussion needs to have the references in the discussion too. For example, I find that the sentences L409-412, L503-507 and L532 need references.

Done.

7. The article adheres to appropriate reporting guidelines and community standards for data availability.

The data and analysis are appropriately reported. However, the authors say that the data are available, but they do not say where.

The program used is now included in the Supporting Information files (see above)

Minor comments

Introduction

L14-18: the sentence is clear, but I suggest cutting this very long sentence

Done

L45: Is there any experimental evidence that individuals indeed use Bayesian updating to estimate their environment?

Psychologists and behavioral ecologists have shown that learning trajectories often follow the patterns predicted by Bayesian models (see also Stamps et al. 2018, cited in this article). However, at this point experimental tests of the predictions of Bayesian models of development over the longer periods of time that are relevant to many questions about the effects of cues on development are still sparse. Our hope is that presenting these models will allow empiricists to test their assumptions and predictions, or at least, modify their experimental designs to control for variation in variables that the Bayesian models suggest are likely to contribute to variation in their results. 

For instance, at present it is rare for empiricists studying TWP to expose parents and offspring to the same cues for the same period of time….but the models suggest that we should expect the results of these experiments to vary as a function of differences in the duration of exposure to the same cues in the two generations. 

L58-69: this paragraph is less clear compared to the rest of the introduction. I suggest a little reorganization:

1) TWP studies focuses on phenotypic values (L58-59)

2) Problem: information-based models tell us mostly about information-states and to have access to phenotypic values on information-based models, you need to include much more factors and assumptions. And if you include many assumptions, you can’t have access to how much information-updating contributes to the phenotypic values (L65-69)

3) the solution is to focus on information-states (L70-71). Not a problem because “when WGP, TGP or TWP occur in response to cues that provide information about the external environment, variation among individuals in their information-states provides the foundation for variation in the trait values observed in experimental studies of these phenomena” (L59-62). Focusing on information-state has already given successful results in other fields (L71-82).

4) What have been done on TWP using information-updating Bayesian models (L83-87)

5) However, currently, nothing known about key factors that impact information-updating (L62-64) and which vary across experimental studies.

Good suggestions.. We have rewritten the relevant sections of the introduction, taking take these comments into account 

L121-126: A small comment but I do not understand why you describe these results so thoroughly. This paragraph is focused on the devaluation of parents’ information compared to offspring’s information, not on the importance of temporal autocorrelation.

OK. We kind of liked the nematode example, because it supported an assumption of the models (that temporal autocorrelation could affect devaluation of information from the parents), but this section is not necessary. So we removed it (and the citations in it) 

L134: it is confusing for a reader to see twice “second” (L113 and L134). I understand after reading again that you want to say that there is two reasons why the assumption L113-155 is not valid. I suggest to say “the first reason is” at the L115 and the “second reason is” at the L134 or to put “third” at the L134 as this sentence refer to the third variable (L91).

Agreed. This section has been rewritten.

L127-133: I would not start a new paragraph, I would do only one paragraph (L113-133), as the ideas of this paragraph (L127-L133) are connected to the previous one.

These topics are connected, but the theoretical models for the first topic (changes in environmental conditions across generations) are different than the models on the second topic (changes within generations)…so we think it is worthwhile to keep them somewhat separate here

L145: little bug? “and/or”

Good catch… thanks

Methods

It is just a suggestion, but I think it will be clearer to see the prior distribution on multiples graphs that you could put in supporting information. I saw after that you did it for your previous paper, which I find very neat.

We have added two new sets of figures to the supporting information files: one set(S1) shows the shapes of the prior distributions used in this study, and a second (S2) shows the shapes of the cumulative likelihood functions used in this study 

L228: little bug, I have a comma that is highlighted in red as when the modifications are tracked in Word.

Fixed…this section has been rewritten

L243: bug, two times “instance”

Fixed

Discussion

L445: bug, a space is highlighted in red.

Fixed. This section has been rewritten

L524-525: this sentence is confusing because with “i.e.”, it looks like you say that species with similar offspring development and parental care share the same constraints of cue degradation. But I think you want to say that closely related species or populations share the same mechanisms to pass on cue, precisely because they are closely related. I would remove “i.e. similar for populations and species with comparable patterns of offspring development and parental care.”

But maybe I did not understand the sentence.

We have rewritten this section, hopefully it is clearer now. 

L539: I am sure you know this, but as stated, it sounds like you think that information-only inductive experiences do not affect the somatic state of parents. Information-only inductive experiences also generate effect on the parental state (e.g. trade-offs between production of defences and other functions, change in life-history strategy), which can have an effect on offspring trait value in addition to the transmission of information from parents to offspring. I would be nice to have a mention of that, otherwise it seems too simplistic (from my point of view).

Of course information-only experiences affect the somatic states of organisms (no changes in the development of phenotypic traits can occur unless the cues affect the physiology or morphology of those organisms). But as is discussed in detail in the cited articles, some types of inductive experiences (e.g. different temperatures) have direct effects on physiology which will occur whether or not the organisms gain any information from those experiences. We have rewritten this section to try to make this clearer.

L533-534: I think you mention in the introduction that it is difficult to link information-state to phenotypic values because of costs of sampling, developmental constraints, or the fitness consequences of expressing different trait values, but this does not appear in the discussion. Even though the point of your paper is to show that information-updating is sufficient to explain the diversity of TWP patterns, I think a few sentences on this topic would be nice to show that it is complicated because these parameters can influence how differences in offspring estimates translate into differences in phenotypic values.

Done…we have rewritten the last paragraph in the article to stress this point

L592-608: just a useless comment but I really like the questions at the end of the discussion dedicated to investigators. I think it is easy to read, instructive and pedagogic.

L596: it would be nice to have some papers or ideas on how estimate the difference in reliability between treatments.

Yes…this is a topic we are currently working on

---

## [Editor Report · Decision Letter 1]

9 Jun 2021

Combining information from parental and personal experiences: simple processes generate diverse outcomes

PONE-D-21-10936R1

Dear Dr. Stamps,

We’re pleased to inform you that your manuscript has been judged scientifically suitable for publication and will be formally accepted for publication once it meets all outstanding technical requirements.

Kind regards,

Peter Schausberger

Academic Editor

PLOS ONE

Additional Editor Comments (optional):

Dear Judy and Alison

Thanks for the swift and thorough revision of your manuscript, which I now happily accept for publication. One minor remark is that I would find it more intuitive to read the acronym TWGP (instead of TWP) for trans- and within-generational plasticity, considering that TGP is used for trans-generational plasticity and WGP for within-generational plasticity.

Best regards

Peter
---

## [Editor Report · Acceptance letter]

5 Jul 2021

PONE-D-21-10936R1 

Combining information from parental and personal experiences:  simple processes generate diverse outcomes 

Dear Dr. Stamps:

I'm pleased to inform you that your manuscript has been deemed suitable for publication in PLOS ONE. Congratulations! Your manuscript is now with our production department. 

Kind regards, 

on behalf of

Dr. Peter Schausberger 

Academic Editor

PLOS ONE